# Effects of Ammonia on Gut Microbiota and Growth Performance of Broiler Chickens

**DOI:** 10.3390/ani11061716

**Published:** 2021-06-08

**Authors:** Hongyu Han, Ying Zhou, Qingxiu Liu, Guangju Wang, Jinghai Feng, Minhong Zhang

**Affiliations:** State Key Laboratory of Animal Nutrition, Institute of Animal Sciences, Chinese Academy of Agricultural Sciences, Beijing 100193, China; hhycaas@163.com (H.H.); 15624955881@163.com (Y.Z.); 18754870532@163.com (Q.L.); 82101185163@caas.cn (G.W.); fjh6289@126.com (J.F.)

**Keywords:** ammonia, gut microbiota, broiler

## Abstract

**Simple Summary:**

The composition and function of gut microbiota is crucial for the health of the host and closely related to animal growth performance. Factors that impact microbiota composition can also impact its productivity. Ammonia (NH3), one of the major contaminants in poultry houses, negatively affects poultry performance. However, the influence of ammonia on broiler intestinal microflora, and whether this influence is related to growth performance, has not been reported. Our results indicated that ammonia caused changes to cecal microflora of broilers, and these changes related to growth performance. Understanding the effects of ammonia on the intestinal microflora of broilers will be beneficial in making targeted decisions to minimize the negative effects of ammonia on broilers.

**Abstract:**

In order to investigate the influence of ammonia on broiler intestinal microflora and growth performance of broiler chickens, 288 21-day-old male Arbor Acres broilers with a similar weight were randomly divided into four groups with different NH3 levels: 0 ppm, 15 ppm, 25 ppm, and 35 ppm. The growth performance of each group was recorded and analyzed. Additionally, 16s rRNA sequencing was performed on the cecal contents of the 0 ppm group and the 35 ppm group broilers. The results showed the following: a decrease in growth performance in broilers was observed after 35 ppm ammonia exposure for 7 days and 25 ppm ammonia exposure for 14 days. At phylum level, the relative abundance of *Proteobacteria* phylum was increased after 35 ppm ammonia exposure. At genus level, ammonia increased the relative abundance of *Escherichia–Shigella* and decreased the relative abundance of *Butyricicoccus*, *Parasutterella*, *Lachnospiraceae_UCG-010*, *Ruminococcaceae_UCG-013* and *Ruminococcaceae_UCG-004.* Negative correlation between *Escherichia–Shigella* and growth performance, and positive correlation between bacteria genera (including *Butyricicoccus*, *Parasutterella*, *Lachnospiraceae_UCG-010*, *Ruminococcaceae_UCG-013* and *Ruminococcaceae_UCG-004*) and growth performance was observed. In conclusion, ammonia exposure caused changes in the structure of cecal microflora, and several species were either positively or negatively correlated with growth performance. These findings will help enhance our understanding of the possible mechanism by which ammonia affect the growth of broilers.

## 1. Introduction

Gaseous ammonia (NH3), the foremost alkaline gas in poultry houses, is emitted primarily by the degradation of litter that contains unutilized nitrogen [1] and negatively affects poultry performance. Studies have shown that a high ammonia environment has toxic effects on organs such as the respiratory tract [2], spleen [3], liver [4], intestine [5] and brain [6]. Additionally, ammonia above 25 ppm reduced body weight gain, feed intake and feed conversion rates on farm animals such as broilers [7,8], laying ducks [9] and pigs [10].

The effects of ammonia, a typical inhaled air pollutant, on gut microbiota has been investigated. In one study, six female pigs were exposed to ammonia (88.2–90.4 mg m^−3^) for 30 days (8 h/day); the results showed that ammonia could cause changes in inflammatory markers and beta diversity of intestinal microflora in fattening pigs [10]. In another study, 60 Shanma ducks were exposed to 75 ppm ammonia for 30 days, the results showed that, the composition of gut microbiota was changed at the phylum and genus levels [11]. Upon broilers’ exposure to ammonia, there are several possible routes of interactions between ammonia and intestinal microflora which may alter intestinal bacterial composition. For example, the gastrointestinal tract is exposed to particles such as PM 2.5, which ammonia plays a significant role in the formation of, through inhalation and ingestion [12]. In addition, ammonia enters the circulatory system from the respiratory tract, which ultimately alters the luminal environment as well as the bacterial composition of the gut [13].

The composition and function of gut microbiota is crucial for the health of the host and closely related to animal growth performance; the relationship between gut microbiota and animal growth performance has also been studied. In one such study, researchers analyzed the cecal and jejunal microbiota of broiler chickens with extreme feed conversion capabilities and found that the abundance of 24 unclassified bacterial species were significantly difference between high and low FCR birds [14]. In another study, the male Cobb broiler chickens, reared in the same conditions, ranked in the top and bottom 12 based on FCR (feed conversion ratio: feed intake/weight gain), AME (Apparent metabolizable energy), GR (weight gain/start weight) and FE (feed consumed) values were selected for microbial analysis, and it was found that *Lactobacillus* correlated with low performance and that *Faecalibacterium* correlated with improved FCR, increased GR and reduced FE [15]. One similar research study in pigs [16] also supports the indication that the gut microbiota is strongly related to the growth performance of animals. 

In order to investigate the influence of ammonia on broiler intestinal microflora and whether this influence related to growth performance, in this study we chose the cecum as a site to investigate the effects of ammonia on broiler intestinal microbiota due to its importance in both metabolism [17] and immune maturation [18]. The results of this research will deepen our understanding of the effects of ammonia in livestock and poultry farms and provide a theoretical basis for promoting poultry intestinal health.

## 2. Materials and Methods

### 2.1. Birds, Diets and Experimental Design

The animal trials were performed as previously described [19]. Briefly, 288 21-day-old male Arbor Acres commercial broilers (Huadu Co., Beijing, China) with a similar weight were randomly divided into 4 treatment groups, each group had 6 replicates (12 broilers per replicate) and each group was housed in a temperature-controlled room. The NH_3_ level of each group was 0 ± 3 ppm, 15 ± 3 ppm, 25 ± 3 ppm and 35 ± 3 ppm, respectively. The chambers were computer-programmed to have the NH_3_ concentration as required. The concentrations of NH_3_ in 4 chambers were monitored with a LumaSense Photoacoustic Field Gas-Monitor INNOVA 1412 (Santa Clara, CA, USA), during the entire experiment. Additionally, airflow was controlled during the exposures to ensure adequate ventilation, minimize build-up of animal-generated contaminants (dander, CO_2_, H_2_S), and avoid thermal stress. The litter material was removed from the chambers every three days to reduce NH3 volatilization. Broiler grower feed and water was supplied ad libitum. The diet composition according to NRC 1994 (National Research Council. 1994. Nutrient Requirements of Poultry) was shown in Table 1.

### 2.2. Sample Collection

All birds were weighted (after 12 h fasting) at 7, 14 and 21 days after the start of the trial, and the performance parameters including ADFI (average daily feed intake), FCR, BWG (body weight gain, calculated as (body weight) − (start weight)) and BW (body weight) were measured. At day 21, one bird of each replication of the 0 ppm group and the 35 ppm group;(a total of 12 birds) were euthanized by cervical dislocation and the cecal contents were obtained and stored at −80 °C for further analyses.

### 2.3. DNA Extraction and PCR Amplification

Microbial DNA was extracted from cecal contents using HiPure Stool DNA Kits (Magen, Guangzhou, China) according to manufacturer’s protocols, and its concentration and quality were determined by the NanoDrop 2000 UV-vis spectrophotometer (Thermo Scientific, Wilmington, NC, USA) and 1% agarose gel electrophoresis, respectively. The V3-V4 hypervariable regions of 16S rRNA gene was amplified with primers 341F (5′-CCTACGGGNGGCWGCAG-3′) and 806R (5′-GGACTACHVGGGTATCTAAT-3′) by the thermocycler PCR system (Qubit 3.0, ThermoFischer Scientific). PCR product was purified using the AMPure XP Beads (Beckman Agencourt, Danvers, MA, USA) and quantified using ABI StepOnePlus Real-Time PCR System (Life Technologies, Foster City, CA, USA). Purified amplicons were pooled in an equimolar solution and then paired-end sequenced (PE250) on an Illumina platform, according to the standard protocols.

### 2.4. Bioinformatic Analysis

Sequencing libraries were generated using SMRTbell TM Template Prep Kit (PacBio, Menlo Park, CA, USA), following manufacturer’s recommendation. The library sequencing was performed on the Illumina HiseqTM 2500/4000 by Gene Denovo Biotechnology Co., Ltd. (Guangzhou, China). According to the unique barcode of each sample, pairing sequences were assigned to each sample, and the primer sequencing and barcode were removed. The matching end was merged and read using FLASH. Based on FASTP (version 0.18.0), quality filtering of raw data was performed under specific filtering conditions to obtain high-quality clean tags. All chimeric tags were removed using the UCHIME algorithm. The feature tables and feature sequences were obtained by DADA2. The clean data was clustered into operational taxonomic units (OTUs) of ≥97% similarity using the UPARSE (version 9.2.64) pipeline. According to the SILVA (release 132) classifier, feature abundance was normalized using the relative abundances of each sample. Alpha diversity was used to analyze the complexity of sample species diversity by Chao1, observed species, goods coverage, Simpson and Shannon, which were calculated using QIIME (version 1.9.1). Beta diversity (between sample), such as PCoA and NMDS, was performed in R project Vegan package (version 2.5.3). The abundance statistics of each taxonomy was visualized using Krona (version 2.6). Species comparison among groups was computed by Tukey’s HSD test in R project Vegan package (version 2.5.3). The function prediction of the OTUs was inferred using PICRUSt (version 2.1.4). Analysis of function difference between groups was calculated by Welch’s *t*-test in R project Vegan package (version 2.5.3).

### 2.5. Statistical Analysis

The data of growth performance were presented as mean ± standard error. The significance of the difference between means was determined by one-way (ANOVA) analysis followed by Tukey’s test, and *p* < 0.05 was considered statistically significant. The correlation between growth performance and microbial species richness was measured by Spearman’s correlation analysis, which was conducted using the R Vegan package.

## 3. Results

### 3.1. Growth Performance

The growth performance of broilers in the different groups was measured by ADFI, F/G, BWG and BW. The results are shown in Table 2. At day 7, F/G was significantly higher (*p* < 0.05) in the 25 and 35 ppm groups compared with the control group, and BWG and BW were significantly lower in the 35 ppm group compared with the other groups. At day 14, ADFI, BWG and BW were significantly lower, and F/G was significantly higher in the 35 ppm group compared with the 0, 15 and 25 ppm groups, and BW in the 25 ppm group was significantly lower compared with the 0 and 15 ppm groups. At day 21, BWG and BW were significantly lower in the 25 and 35 ppm groups compared with the 0 and 15 ppm groups, and BWG and BW in the 35 ppm group were significantly lower compared with the 25 ppm group. Moreover, ADFI and F/G were significantly lower and higher separately in the 35 ppm group compared with the other groups.

### 3.2. Summary of Sequencing Data

A total of 1,525,416 clean tags were obtained and clustered into 13,984 OTUs (1075.75 ± 73.48 per sample); the detailed information of sequencing data was shown in Figure 1. There were 264 and 196 unique OTUs in the control group and the treatment group (the 35 ppm ammonia group, for ease of description, was labeled as treatment group, TG), respectively, and 709 OTUs were the same between the two groups, as shown in Figure 2A.

### 3.3. Alpha and Beta Diversity Analysis

The effects of ammonia on alpha diversity in the cecal contents of broilers is shown in Table 3. The rarefaction curves of observed species showed a flattening trend in Figure 2B, indicating enough sequence depth to cover almost overall sequences; that is, almost all bacterial species are present in the cecal contents of broilers. Whether Shannon or Simpson indices reflect community species diversity or Chao1, ACE indices reflect that richness was not significantly different among the two groups (*p* > 0.05).

PCoA (Principal Coordinate Analysis) based on Unweighted-UniFrac and NMDS (Non-metric Multi-Dimensional Scaling) were used to visualize the beta diversity of cecal microbiota from two groups. Analysis of similarities based on unweighted UniFrac is shown in Figure 3. There were significant differences in beta diversity among different groups (R = 0.287, *p* < 0.05).

### 3.4. Analysis of Cecal Microbiota at Phylum and Genus Level

A taxonomic analysis of the gut microflora in the cecal contents of broiler was shown in Figure 4. At phylum level, *Firmicutes* and *Bacteroidetes* phyla acted as the predominant taxa in two groups. The *Proteobacteria* phylum relative abundance in the treatment group was significantly higher (*p* < 0.05) compared with the control group. At genus level, there were significant differences (*p* < 0.05) in six microbial species between the treatment group and control group, among which the relative abundance of *Escherichia–Shigella* was higher (*p* < 0.05) in the treatment group, and the relative abundance of *Butyricicoccus*, *Parasutterella*, *Lachnospiraceae_UCG-010*, *Ruminococcaceae_UCG-013* and *Ruminococcaceae_UCG-004* was lower in the treatment group.

### 3.5. Function Prediction

The differences of presumptive biological function of cecal contents microflora between the control group and the treatment group were presented in Figure 5. The metabolic pathway associated with amino acids, cofactors and vitamin metabolism, replication and repair, signal transduction, transport and catabolism was decreased in the treatment group (*p* < 0.05).

### 3.6. Correlation of Intestinal Microflora and Growth Performance

The correlation between growth performance and indicator species was shown in Figure 6. The results shown that *Escherichia–Shigella* was positively correlated with F/G and negatively correlated with ADFI, BWG and BW. *Ruminococcaceae-UCG-013* was negatively correlated with F/G and positively correlated with BWG and BW. *Lachnospiraceae-UCG-010* was positively correlated with ADFI, BWG and BW, and negatively correlated with F/G. *Parasutterella* was positively correlated with ADFI, BW and BWG; *Ruminococcaceae-UCG-004* and *Butyricicoccus* were positively correlated with ADFI, BWG and BW.

## 4. Discussion

In this study, the effects of ammonia on growth performance and the intestinal microflora of broiler chickens were investigated. Additionally, the correlation between growth performance and intestinal microflora under ammonia exposure has also been analyzed.

### 4.1. The Effect of Ammonia on Growth Performance of Broiler Chickens

Ammonia is the most prominent toxic gas in poultry houses. It originates from the breakdown of undigested protein and uric acid in the litter [20] and adversely affects the health [21,22] and growth performance [7,8,23,24] of broilers. In this experiment, a decrease in growth performance was observed in the 25 ppm ammonia group and the 35 ppm ammonia group. Our results are consistent with previous studies [7,8,25].

### 4.2. The Effects of Ammonia on Intestinal Microflora

Our results showed that a variety of microbiota communities lived in the cecal of broilers, and there was a significant difference in gut microflora composition and structure among two groups, indicating that ammonia exposure could induce changes in the symbiotic relationship of microorganisms. Ammonia exposure changed the beta diversity but did not change the alpha diversity, indicating that ammonia changes the species composition but hardly influences the species richness and evenness. The structural response of gut microbiota to ammonia exposure was investigated in this study; results showed that *Firmicutes*, *Bacteroidetes* and *Proteobacteria* were the most dominant phyla in the cecal of broilers between two groups, which was consistent with previous studies [26,27,28]. Although the abundance of the first two was no significant difference between the two groups, the abundance of *Proteobacteria* was higher in the ammonia group, which is in agreement with the increase of *Escherichia–Shigella* genus.

Some species of *Escherichia–Shigella* genera invade and proliferate within colonocytes and mucosal macrophages [29], while the ammonia often leads to abnormal intestinal structure and dysfunction. It was reported that the villus height and crypt depth in the small intestinal was much lower in broilers after 75 ppm ammonia exposure [30]. Damages of gut structure caused by ammonia possibly promote the proliferation of *Escherichia–Shigella*. 

Simultaneously, the significantly lower relative abundance of *Butyricicoccus*, *Parasutterella*, *Lachnospiraceae_UCG-010* and *Ruminococcaceae_UCG-013*, *Ruminococcaceae_UCG-004* was observed in the ammonia group. *Butyricicoccus*, *Ruminococcaceae_UCG-013* and *Ruminococcaceae_UCG-004* belong to the same phylogenetic order (Clostridiales), which is autochthonous benign microbes primarily inhabit the caecum [31], which is the predominant SCFAs-producing bacteria in the gut [32]. In addition, *Lachnospiraceae* benefits gut development and health by degrading plant fiber and producing SCFAs [33]. These types of bacteria prefer an acidic intestinal environment; however, previous studies found that ammonia exposure results in increased blood ammonia concentration in broilers [6,8], which could induce the increase in intracellular and extracellular pH [34]. All of this inhibited the proliferation of acid-producing bacteria.

### 4.3. Correlation between Growth Performance and Intestinal Microflora

The composition and function of gut microbiota is crucial for the health of the host and closely related to animal growth performance. The relationship between intestinal flora and growth performance was explored by observational experiments in previous studies and, indeed, there was a significant difference in gut microflora composition and structure between the birds with the highest and lowest feed conversion efficiency [14,15,35]. 

In this study, we analyzed the relationship between growth performance and the different genera that ammonia caused. A negative correlation between *Escherichia–Shigella* and ADFI, BWG and BW was observed, which is in line with previous studies [36]. Another study also linked the lower proportion of enterobacteria with improved performance in broilers [37]. The hyperproliferation of enterobacteria in the gut resulted in mucosal impairment, villus erosion and damage to the intestinal cells [38], and caused intense acute inflammatory response and cytokines release in intestinal tissue [29], thus reducing its nutrients’ absorptive potential. This may explain, at least in part, the negative correlation between enterobacteria and growth performance.

In this study, the relative abundance of *Butyricicoccus, Parasutterella, Ruminococcaceae_UCG-004, Ruminococcaceae-UCG-013* and *Lachnospiraceae-UCG-010* was positively related with growth performance of broilers. Except for the *Parasutterella,* all of these genera were related to the order Clostridiales, which ferments diverse plant polysaccharides to SCFAs [39]. In line with this result, Stanley [14] found that the closest blast matches to three potentially performance beneficial OTUs were related to the order Clostridiales.

SCFAs, the major component constitutes of bacterial metabolites, have broad effects on various aspects of host physiology [40], especially against intestinal inflammation and in maintaining intestinal epithelial integrity [41]. For example, butyrate not only stimulates anti-inflammatory pathways in the gut [42], but also increases the expression of tight junction proteins and promotes proliferation of normal intestinal epithelial cells [43].

The genus of *Parasutterella* has been defined as a core component of gut microbiota, correlated with various microbial-derived metabolites such as aromatic amino acid, bilirubin, purine and bile acid derivatives [44]. Alteration of microbially derived metabolites is an important mechanism through which changes in gut microbial activity generate functional consequences for host health outcomes [45,46]. In this study, the lower relative abundance of *Parasutterella* in the treatment group is in keeping with the depleted amino acid, cofactors and vitamin metabolism pathways, together with poor performance.

## 5. Conclusions

In conclusion, in this study, we found that ammonia exposure caused changes in the structure of cecal microflora. At the genus level, we found that several species were positively or negatively correlated with growth performance. These results will help enhance our understanding of the possible mechanism by which ammonia affects the growth of broilers.

## Figures and Tables

**Figure 1 animals-11-01716-f001:**
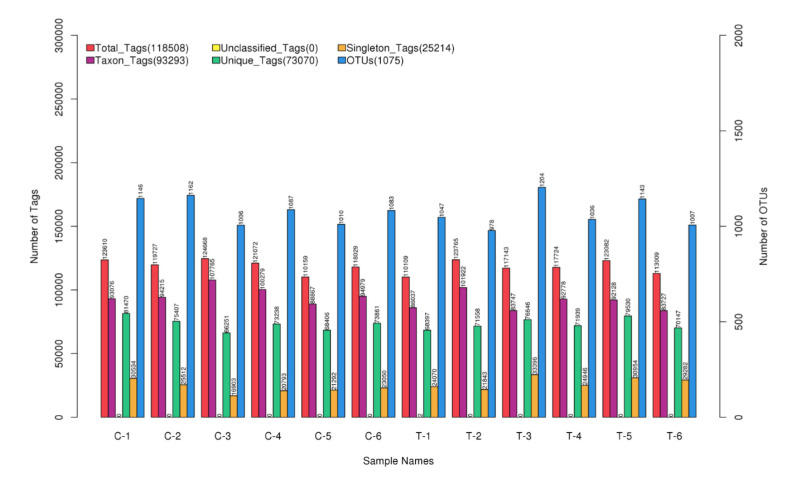
Statistics of the number of OTUs and Tags of different samples.

**Figure 2 animals-11-01716-f002:**
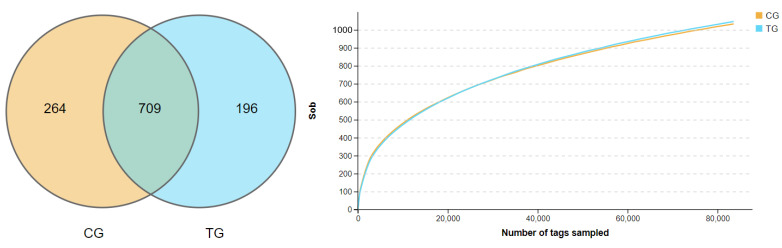
(**A**) Venn maps of OTUs. (**B**) The rarefaction curves of observed species. CG (control group), TG (treatment group of 35 ppm ammonia).

**Figure 3 animals-11-01716-f003:**
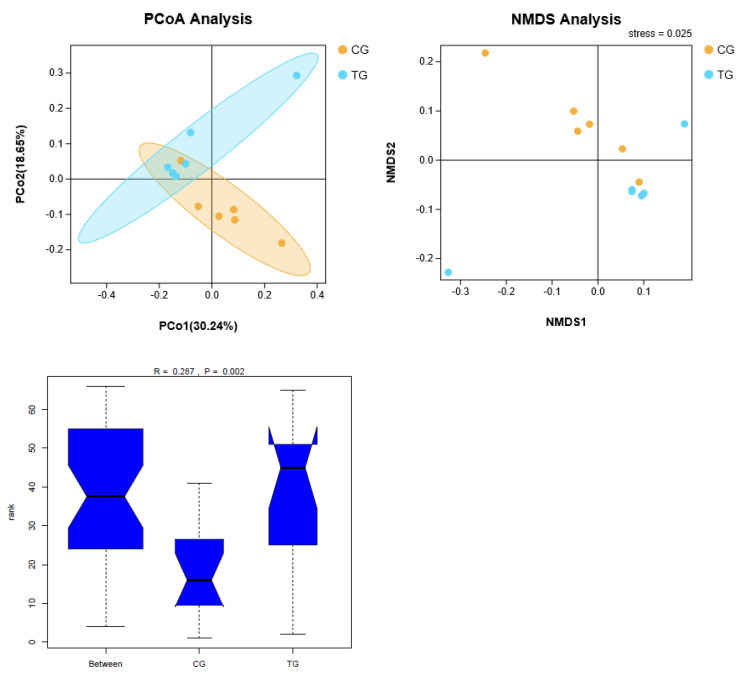
The effects of NH_3_ on beta diversity of cecal microbiota in broiler chickens. (**A**) PCoA: Principal coordinates analysis; (**B**) NMDS: Non-metric Multi-Dimensional Scaling. (**C**) Analysis of similarities.

**Figure 4 animals-11-01716-f004:**
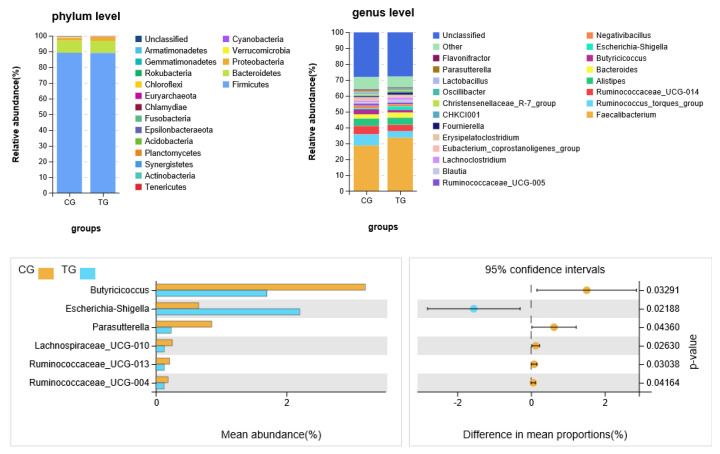
Accumulation map of broiler cecal microorganism abundance at the (**A**) phylum and (**B**) genus level. (**C**) Six bacteria with significant difference in relative abundance between two groups.

**Figure 5 animals-11-01716-f005:**
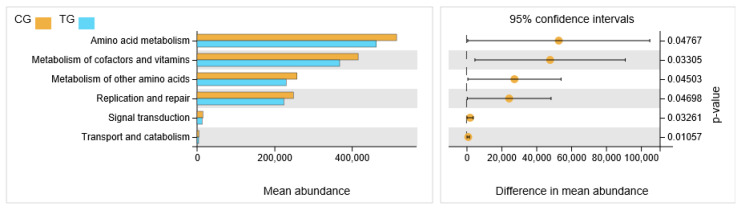
The differences of presumptive biological function of cecal contents microflora between the control group and the treatment group.

**Figure 6 animals-11-01716-f006:**
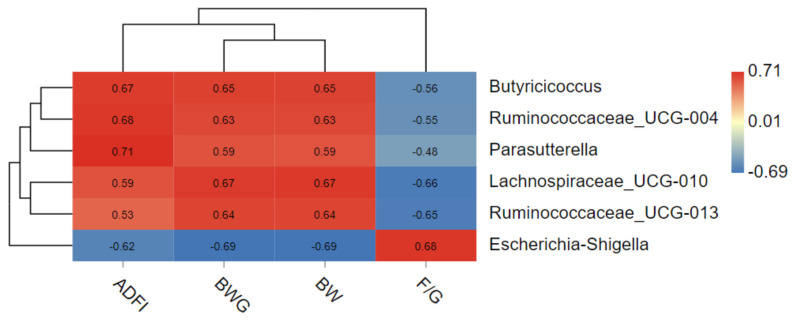
Correlation analysis of growth performance (ADFI, BWG, BW, F/G) and 6 indicator species.* 0.01 < *p* ≤ 0.05.

**Table 1 animals-11-01716-t001:** Ingredients and nutrient compositions of the basal diet.

Ingredients (g/kg)	Content (%)
Corn	56.51
Soybean meal	35.52
Soybean oil	4.50
NaCl	0.30
Limestone	1.00
Dicalcium phosphate	1.78
DL-Methionine	0.11
Premix ^1^	0.28
Total	100.00
Calculated nutrient levels	
Metabolizable energy (MJ/kg)	12.73
Crude protein (g/kg)	20.07
Available Phosphorus (g/kg)	0.40
Calcium (g/kg)	0.90
Lysine (g/kg)	1.00
Methionine (g/kg)	0.42
Methionine + cysteine (g/kg)	0.78

^1^ Premix provided the following per kg of the diet: vitamin A, 10,000 IU; vitamin D3, 3400 IU; vitamin E, 16 IU; vitamin K3, 2.0 mg; vitamin B1, 2.0 mg; vitamin B2, 6.4 mg; vitamin B6, 2.0 mg; vitamin B12, 0.012 mg; pantothenic acid calcium, 10 mg; nicotinic acid, 26 mg; folic acid, 1 mg; biotin, 0.1 mg; choline, 500 mg; Zn (ZnSO_4_·7H_2_O), 40 mg; Fe (FeSO_4_·7H_2_O), 80 mg; Cu (CuSO_4_·5H_2_O), 8 mg; Mn (MnSO_4_·H_2_O), 80 mg; I (KI) 0.35 mg; Se (Na_2_SeO_3_), 0.15 mg.

**Table 2 animals-11-01716-t002:** Growth performance indices of AA broilers in different groups.

Time Points	Treatment	ADFI/g	F/G	BWG/kg	BW/kg
0–7	0 ppm	104.91 ± 3.08	1.31 ± 0.04 ^c^	0.48 ± 0.18 ^a^	1.173 ± 0.008 ^a^
	15 ppm	108.48 ± 2.16	1.35 ± 0.04 ^c^	0.45 ± 0.36 ^a^	1.173 ± 0.008 ^a^
	25 ppm	107.28 ± 2.36	1.44 ± 0.05 ^b^	0.48 ± 0.20 ^a^	1.142 ± 0.015 ^a^
	35 ppm	101.52 ± 1.44	1.53 ± 0.03 ^a^	0.40 ± 0.29 ^b^	1.092 ± 0.017 ^b^
	*p*-value	0.1927	0.0047	<0.0001	0.0004
0–14	0ppm	115.09 ± 2.47 ^a^	1.50 ± 0.03 ^b^	0.99 ± 0.03 ^a^	1.686 ± 0.014 ^a^
	15 ppm	114.25 ± 1.74 ^a^	1.58 ± 0.05 ^b^	0.90 ± 0.05 ^a^	1.634 ± 0.027 ^a^
	25 ppm	114.25 ± 2.33 ^a^	1.65 ± 0.06 ^b^	0.95 ± 0.07 ^a^	1.600 ± 0.021 ^b^
	35 ppm	96.47 ± 1.18 ^b^	1.87 ± 0.06 ^a^	0.67 ± 0.04 ^c^	1.365 ± 0.017 ^c^
	*p*-value	<0.0001	0.0004	<0.0001	<0.0001
0–21	0 ppm	119.55 ± 2.27 ^a^	1.57 ± 0.02 ^b^	1.52 ± 0.05 ^a^	2.216 ± 0.021 ^a^
	15 ppm	118.19 ± 1.55 ^a^	1.60 ± 0.02 ^b^	1.47 ± 0.06 ^a^	2.166 ± 0.019 ^a^
	25 ppm	118.62 ± 2.38 ^a^	1.70 ± 0.02 ^b^	1.39 ± 0.05 ^b^	2.091 ± 0.026 ^b^
	35 ppm	99.52 ± 1.40 ^b^	2.41 ± 0.08 ^a^	0.83 ± 0.05 ^c^	1.522 ± 0.026 ^c^
	*p*-value	<0.0001	<0.0001	<0.0001	<0.0001

^a^, ^b^, ^c^ Means with different letters in the same column indicated a significant difference (*p* < 0.05).

**Table 3 animals-11-01716-t003:** *p*-value in α-diversity indices between control group and treatment group.

Index	Control Group	Treatment Group	*p*-Value
sobs	1082.3 ± 65.55	1069.1 ± 86.48	0.77
Shannon	5.79 ± 0.44	5.43 ± 0.43	0.18
Simpson	0.91 ± 0.05	0.88 ± 0.05	0.41
Chao1	1499.8 ± 71.68	1420.6 ± 133.11	0.23
ACE	1530.18 ± 68.63	1486.13 ± 104.39	0.49
Good’s coverage	0.996	0.996	0.52

## Data Availability

Not applicable.

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
