# Peer review of "Effects of Ammonia on Gut Microbiota and Growth Performance of Broiler Chickens"

_animals, 2021, doi:10.3390/ani11061716_

Round 1
Reviewer 1 Report
The authors tried to determine the effects of ammonia on the gut microbiota and physical conditions in broilers. The topic is highly interesting and of scientific significance but it is flawed by very weak English which needs to be improved. Besides few comments as below:
Line 60: replace eaten by consumed
Section 2.1: As far as I understand, the authors used 288 birds in four dietary groups of each six replicates which further had 12 replicates each. If it is true-please justify that the number of the birds were sufficient for this experiment and no excessive number of birds were utilized/slaughtered?
Line 90: This sentence is not clear and is not easy to understand.Â
Section 2.2: Please mention clearly how many number of birds were slaughtered from each group and how many number of samples were processed for DNA extraction and sequence analysis. Also mention the total number of samples to determine the mean of each body weight value describe in the whole manuscript.Â
Sections 2.3 and 2.4: The heading is the same
Section 3.6: remove e word significantly from the whole paragraph.Â
Conclusions needed to be rewritten. It is not understandable to take a message form the current form.
Reviewer 2 Report
Introduction
- The introduction does not fully clarify how ammonia inhaled through the respiratory tract can affect the gut microbiota and what the link is between the respiratory and gastrointestinal tracts.
- The aim of the study is not clearly stated.
Materials and methods
- The method of determining the ammonia concentration in the chickens and maintaining the continuity of a given concentration in each group is not described.
- Why were the birds sacrificed only in the control group and the group in which the ammonia concentration was 35 ppm? Why wasn’t the same determined in the other groups?
- The sampling of the intestines is not precisely described. Were only aerobic bacteria analysed?
Discussion
- If the effect of ammonia on the growth of chickens is known, as the authors state, then this stage of the experiment is unnecessary and should not have been included in the study.
- There is no specific analysis of why ammonia affects the bacterial population.
Reviewer 3 Report
The authors investigated the effects of ammonia on gut microbiota and the growth performance of broiler chickens. This research is unique in that it evaluated the effect on gut microbiota, however, it has major issues on justification/Intro, M&M section, and interpretation of results. The specific comments are mentioned below.
Line 34-37: Please rewrite these sentences to improve clarity.
Line 64-66: Although the study on the effect of ammonia on gut microbiota is very limited, there is literature about the effect of ammonia on the growth and health of birds. I suggest the author add more info about it. In addition, please add more info for the justification of this study.
Â
Line 76: What is the level of ammonia in commercial poultry production systems in broiler and layers?
Line 77: How did you achieve and maintain the desired ammonia levels for the research purpose? Since the ammonia concentration varies with the age of broilers, how frequently did you measure the levels to ensure that it lies within the range of tested concentration? Also please add info regarding litter material and it changes the NH3 levels.Â
Line 93-104 and 105-115: Please delete repeating information.
Line 236-238: Please avoid repeating the result in the discussion section and instead focus on discussing the results.
Line 243: Discuss the results of alpha and beta diversity. What could be the reason for not observing the difference in alpha diversity?
Â
Â
Â
Â
Â
Round 2
Reviewer 1 Report
Few English mistakes including grammatical errors, numbering and heading/subheading cross check. Please try to add the references you mentioned in the address to the comments.
Reviewer 2 Report
none
Reviewer 3 Report
Thank you, authors, for the revisions. I still believe it needs major revision because.
- The data of the cecal microbiome must be re-analyzed since the concentration of NH3 could impact the gut microbiota differently. The authors had grouped all the treatments into a group (labeled as TG) and compared it with control (Line 176, Fig 4) and this will mislead the reader for understanding the effect of different concentration of ammonia on the gut microbiome especially since the main aim of this research is to understand the effect of NH3 on the gut microbiome.
- For the statistical analysis, I suggest using multiple comparison tests for the effect of different concentrations of ammonia on the cecal microbiome. Please add more information, how the OTU was identified, what level of similarity, and which software was used.
- The author should revise the results and discussion to present the effect of different levels of concentration of ammonia on the gut microbiome after re-analysis of data.
Round 3
Reviewer 3 Report
Thank you for the clarification and for adding the details to the manuscript.Â